# Cherry Hypothesis 🍒: Identifying the Cherry on the Cake for Dynamic Networks

## Abstract

Dynamic networks, e.g., Dynamic Convolution (DY-Conv) and the Mixture of Experts (MoE), have been extensively explored as they can considerably improve the model's representation power with acceptable computational cost. The common practice in implementing dynamic networks is to convert given static layers into fully dynamic ones where all parameters are dynamic (at least within a single layer) and vary with the input. Recent studies empirically show the trend that the more dynamic layers contribute to ever-increasing performance. However, such a fully dynamic setting 1) may cause redundant parameters and high deployment costs, limiting the applicability of dynamic networks to a broader range of tasks and models, and more importantly, 2) contradicts the previous discovery in the human brain that *when human brains process an attention-demanding task, only partial neurons in the task-specific areas are activated by the input, while the rest neurons leave in a baseline state.* Critically, there is no effort to understand and resolve the above contradictory finding, leaving the primal question – whether to make the computational parameters fully dynamic or not? – unanswered. The main contributions of our work are challenging the basic commonsense in dynamic networks, and, proposing and validating the CHERRY HYPOTHESIS – *A fully dynamic network contains a subset of dynamic parameters that when transforming other dynamic parameters into static ones, can maintain or even exceed the performance of the original network.* Technically, we propose a brain-inspired partially dynamic network, namely PAD-Net, to transform the redundant dynamic parameters into static ones. Also, we further design Iterative Mode Partition to partition the dynamic- and static-subnet, which alleviates the redundancy in traditional fully dynamic networks. Our hypothesis and method are comprehensively supported by large-scale experiments with two typical advanced dynamic methods, i.e., DY-Conv and MoE, on both image classification and GLUE benchmarks. Encouragingly, we surpass the fully dynamic networks by +0.7% top-1 acc with only 30% dynamic parameters for ResNet-50 and +1.9% average score in language understanding tasks with only 50% dynamic parameters for BERT-base. As for reproducibility, the code has been uploaded to OpenReview and will be released upon acceptance.

## 1 Introduction

In past years, deep neural networks have been continuously pushing the state-of-the-art performance in the tasks of computer vision (Girshick et al., 2014; Dosovitskiy et al., 2021) and natural language processing (Dai & Le, 2015; Brunet et al., 2019; Zhong et al., 2022a). However, most prevalent architectures perform inference in a static manner where both the computational graph and network parameters are fixed once after training, which limits the representation power. Dynamic networks (Han et al., 2021), as opposed to static ones, adapt their parameters or architectures to each specific input, improving the model representation power with acceptable computational cost, e.g., Switch Transformers (Fedus et al., 2021). The common practice of implementing dynamic networks is transforming static networks (or modules) with counterpart dynamic ones, for example, Dynamic Convolution (Chen et al., 2020b) replaces traditional convolution by adopting $k$ additive convolutional kernels; Mixture of Experts (Shazeer et al., 2017) replaces a fully connected layer with multiple feed-forward neural networks (FFNs) in parallel.

Previous works (Chen et al., 2020b; Li et al., 2021a) show that dynamic networks often outperform their static counterpart, and using more dynamic layers intriguingly leads to ever-increasing performance. For instance, dynamic convolution increasingly promotes the performance on the ImageNet when more static convolution layers turn into dynamic ones. However, these basic settings in dynamic networks contradict the phenomenon that neuroscience researchers observed in human brains. A notable amount of neuroscience research efforts (Hamilton et al., 2011; Raichle, 2015) has been conducted to reveal that a subnetwork of the human brain remains in a baseline state (like static networks) instead of being activated when people tackle attention-demanding tasks like watching movies (Hasson et al., 2008) or reading books (Lerner et al., 2011), which are named as the task-negative network (TNN) (Raichle et al., 2001) in neuroscience. Whilst another subnetwork that is activated by the input (like dynamic networks) is named the task-positive network (TPN) (Ptak, 2012). This raises a seemingly counterintuitive contradiction that the discovery in the human brain is inapplicable to the dynamic networks. Critically, there is no effort to understand and resolve this seeming contradiction. Consequently, for practitioners, it remains unclear whether to make the computational parameters dynamic and to what extent if yes.

We answer the above questions by meticulously revisiting the properties of dynamic networks. Existing dynamic networks often follow a fully dynamic manner, where all parameters are dynamic (at least within a single layer) and vary with the input. Such a fully dynamic manner is resource expensive and may cause redundancy, limiting the applicability of dynamic networks. For instance, the total parameters of ResNet-50 equipped with dynamic convolution are ~100.9M (with 4 kernels) compared to only ~23.5M for vanilla ResNet-50. It seems more is better when transforming static layers into dynamic ones, but how about the dynamic parameters within a dynamic network: Are all of them cherries that lead to the promotion? This urges us to reflect (1) *Whether there exist redundant dynamic parameters, in the fully dynamic network layers?* In addition, other studies (Fox et al., 2005; Damoiseaux et al., 2006) reveal that TPN and TNN intermingle and coexist in regions of the brain, including the temporal lobe and prefrontal lobe, and angular convolution, making us reflect (2) *Whether the coexistence of dynamic and static parameters brings better effects for the fully dynamic network layers?* Based on the above scrutinization, we assume that less can compete with more for dynamic parameters in fully dynamic networks.

Formally, we cautiously propose the **Cherry Hypothesis**: *A fully dynamic network 🍰 contains a subset of dynamic parameters 🍒 that when transforming other dynamic parameters into static ones, can maintain or even exceed the performance of the original network.*

With this hypothesis, we propose the Iterative Mode Partition (IMP) algorithm to transform less important dynamic parameters into static ones step by step, expecting competitive performance with higher efficiency. Given a fully dynamic network initialized with all parameters in dynamic mode, we attempt to partition a subset of static mode parameters out from them. Specifically, we iteratively transform dynamic parameters and measure the influence on the loss values. If the transformation of the $i$-th element of the dynamic parameters only causes a minimal difference in the loss values, we can safely transform this parameter into a static one. Given the desired dynamic ratio (the proportion of dynamic parameters), we can balance the trade-off between dynamic and static parameters. After mode partition, for a few dynamic parameters to reserve and generate, we can prune out redundant parameters and obtain a light-weight architecture, namely Partially Dynamic Networks (**PAD-Net**), which contains two modes of parameters (dynamic parameters that vary with specific inputs and static parameters that is fixed during inference).

Empirically, we extensively validate the cherry hypothesis and our proposed PAD-Net, including visual image classification (Deng et al., 2009) for dynamic convolution and GLUE benchmark (Wang et al., 2019) for MoE. Experiment results reveal that we succeed in transforming redundant dynamic parameters into static ones and our proposed model PAD-Net achieves the highest performance in all tasks with lightweight architectures. Given the superiority of PAD-Net in both effectiveness and efficiency, we show that less dynamic is more efficient in fully dynamic networks, successfully verifying the cherry hypothesis. The inspiration of partially dynamic can be extended to other dynamic networks and even inform future efficient architectures designation.

In short, our contributions are threefold:

- We give the brain-inspired Cherry Hypothesis for the existing dynamic networks to identify the sub-network that maintains or exceeds the representation power of the original fully dynamic networks.

- Following our hypothesis, we propose the novel PAD-Net to achieve the mode partition, where an *Iterative Mode Partition* (IMP) algorithm is designed to partition the parameters into two modes.

- We empirically validate our hypothesis and PAD-Net on both CV and NLP tasks across two representative dynamic networks, including Dynamic Convolution and Mixture of Experts.

## 2 Related Work

**Dynamic Networks.** The dynamic neural network is an emerging research topic in deep learning, which adapts structures or parameters to different inputs, leading to notable advantages in terms of accuracy, and computational efficiency. Han et al. (2021) classify dynamic networks into two categories: dynamic architecture networks and dynamic parameter networks. Dynamic architecture networks perform inference with specific architectures conditioned on each sample. Specifically, they adaptively adjust the network depth (Wang et al., 2018), width (Mullapudi et al., 2018), or route based on the input (Huang et al., 2018). Instead of changing the model architecture, dynamic parameter networks boost representation power by adapting parameters or activation functions to the input (Yang et al., 2019; Liu et al., 2021). Existing works often transform various types of static parameters into dynamic versions (Chen et al., 2020b). Among them, dynamic convolution (and mixture-of-experts) are the typical examples that aggregate multiple convolution parameters (and experts) dynamically based on the input, leading to significant improvement with negligible computational cost.

**The Task-Negative Network.** Neural networks originate from the success of the human brain in neuroscience (Eslami et al., 2016; Higgins et al., 2016). Recent studies (Di & Biswal, 2014; Cheng et al., 2020) reveal that a mature brain works as a unified system involving various functionally specialized networks that deal with an array of particular functions called functional networks, including the task-negative network (TNN) and the task-positive network (TPN) (Hamilton et al., 2011). TNN shows decreased activity (or static) during the performance of attention-demanding tasks, while TPN exhibits increased activity (or dynamic) during the execution of an attention-demanding task (Fransson, 2005). Our motivation for adopting two subnets of parameters to fully dynamic networks is to borrow the functionality of TPN (while leaving the rest as TNN) in neuroscience.

**Network Pruning.** Past works in network pruning have explored effective techniques to find efficient subnetworks (Lee et al., 2019; Evci et al., 2020; He et al., 2022; 2023) and zero out redundant parameters. According to the lottery ticket hypothesis (LTH) pioneered by Frankle & Carbin (2019), dense, randomly initialized, feed-forward networks contain the subnetwork (winning tickets) that maintains comparable test performance of the original network after training for the same iterations. This hypothesis inspires a series of follow-up works in network pruning. However, these methods always sacrifice performance because of pruned parameters. Instead of directly pruning the dynamic parameters in dynamic networks, we considered changing them to static ones. In Section 5.3, we show that our approach significantly and consistently outperforms fully dynamic networks in the GLUE benchmark (Wang et al., 2019), while the pruned model performed worse than the original network.

## 3 Preliminaries: Review of Fully Dynamic Networks

**Basic Concept.** Dynamic networks adopt an indirect way to compute with the input, where the network first adjust the computational parameters and then utilizes them to compute with the input, instead of directly taking the intrinsic parameters as computational parameters. In fully dynamic networks, intrinsic parameters are used as dynamic factors that participate in the generation of computational parameters, and the computational parameters $\hat{\Theta}$ are based on two parts: the input $\mathbf{x}$ and the intrinsic parameters $\Theta$. Let us denote $\mathcal{W}$ as the dynamic function for producing the computational parameters and the computational parameters can be written as $\hat{\Theta} = \mathcal{W}(\mathbf{x}, \Theta)$. Given an input sample $\mathbf{x}$, the output of a conventional network

with static parameters can be $\mathbf{y} = \mathcal{F}(\mathbf{x}, \Theta)$. In dynamic networks, this equation can be reformulated as $\mathbf{y} = \mathcal{F}(\mathbf{x}, \hat{\Theta}|\mathbf{x})$, where the difference lies in whether the computational parameters are based on the input.

Although using different dynamic functions, existing dynamic networks often follow a fully dynamic manner: Networks take all intrinsic parameters to generate the computational parameters where all elements are dynamic and vary with the input. We call such networks fully dynamic networks and, in the following, introduce instantiations coming from dynamic parameter networks, i.e., ***Dynamic Convolution***, and dynamic architecture networks, i.e., ***Mixture of Experts***, respectively.

**Dynamic Convolution.** As a typical example of dynamic parameter networks, Dynamic Convolution (Chen et al., 2020b) prepares $k$ parallel static kernels $\Theta^{(i)}(i = 1, 2, \ldots, k)$ as intrinsic parameters and utilizes the linear combination of them as the aggregated kernel. The linear scale is aggregated dynamically via a channel-wise attention block (Hu et al., 2018) denoted as Attention, the dynamic function of dynamic convolution can be written as:

$$\mathcal{W}(\mathbf{x}, \Theta) = \sum_{i=1}^{k} \pi_i(\mathbf{x}) \cdot \Theta^{(i)}, \quad \text{where} \quad \pi(\mathbf{x}) = \text{Attention}(\mathbf{x}). \tag{1}$$

**Mixture of Experts.** We talk about dynamic architecture networks by taking the Mixture of Experts (MoE) (Jacobs et al., 1991; Shazeer et al., 2017) as an instantiation. MoE prepares $m$ parallel static experts $\Theta^{(i)}(i = 1, 2, \ldots, k)$ and only selects $n(n < m)$ experts with the highest scores. Given a specific input, we denote $G(\mathbf{x})$ as the output scores of gating and $\mathcal{T}$ as the indices of the selected experts. For the $i$-th selected expert, we denote the combination of the score $G_{\mathcal{T}_i}(\mathbf{x})$ and parameters $\Theta^{(\mathcal{T}_i)}$ as $w^{(\mathcal{T}_i)} = \{G_{\mathcal{T}_i}(\mathbf{x}), \Theta^{(\mathcal{T}_i)}\}$. The dynamic function of MoE can be represented as:

$$\mathcal{W}(\mathbf{x}, \Theta) = \{w^{(\mathcal{T}_1)}, \ldots, w^{(\mathcal{T}_n)}\}, \quad \text{where} \quad w^{(\mathcal{T}_i)} = \{G_{\mathcal{T}_i}(\mathbf{x}), \Theta^{(\mathcal{T}_i)}\}. \tag{2}$$

**Limitation Discussions.** Mainstream dynamic networks usually replace static layers with fully dynamic layers. In these layers, all elements of dynamic parameters require counterpart dynamic factors co-working with the input sample. However, this situation may cause redundant parameters and high deployment costs, limiting the applicability of dynamic networks to a border range of resource-constrained situations and large-scale models. For this fully dynamic manner, we raise two questions: (1) *Is it necessary to pay the cost of enormous parameters and computations, to aggregate the dynamic parameters?* (2) *Is it necessary to make all computational parameters dynamic to maintain the slight performance improvement?* Inspired by the co-working of task-negative networks (TNN) and task-positive networks (TPN), we propose the Partially Dynamic Network (PAD-Net) that mixes dynamic parameters and static parameters to answer the above questions.

## 4 Methodology

To overcome the aforementioned challenges and limitations, we propose a novel network architecture, Partially Dynamic Network (PAD-Net). We also devise a new algorithm *Iterative Mode Partition* (IMP) to build this model efficiently.

### 4.1 PAD-Net: Partially Dynamic Network

In response to the limitation of fully dynamic networks, we question whether it is necessary to make all parameters dynamic. To this end, we try to detect the less important dynamic parameters and transform them into input-agnostic static parameters. Specifically, we utilize a mask $M_i(i = 1, 2, \ldots, m)$ to indicate whether the $i$-th element of $\hat{\Theta}$ is in dynamic or static mode: $M_i = 1$ means the $i$-th element of $\hat{\Theta}$ is dynamic and vice versa. Considering two modes of parameters in our model, we use $\tilde{\Theta} \in \mathbb{R}^m$ to denote the dynamic parameters and adopt additional parameters $\bar{\Theta} \in \mathbb{R}^m$ to represent the static parameters, then the computational parameters $\hat{\Theta}$ can be formulated as:

$$\hat{\Theta}_i = \begin{cases} \tilde{\Theta}_i = \mathcal{W}_i(\mathbf{x}, \Theta) & \text{if } M_i = 1 \\ \bar{\Theta}_i & \text{otherwise} \end{cases}, \tag{3}$$

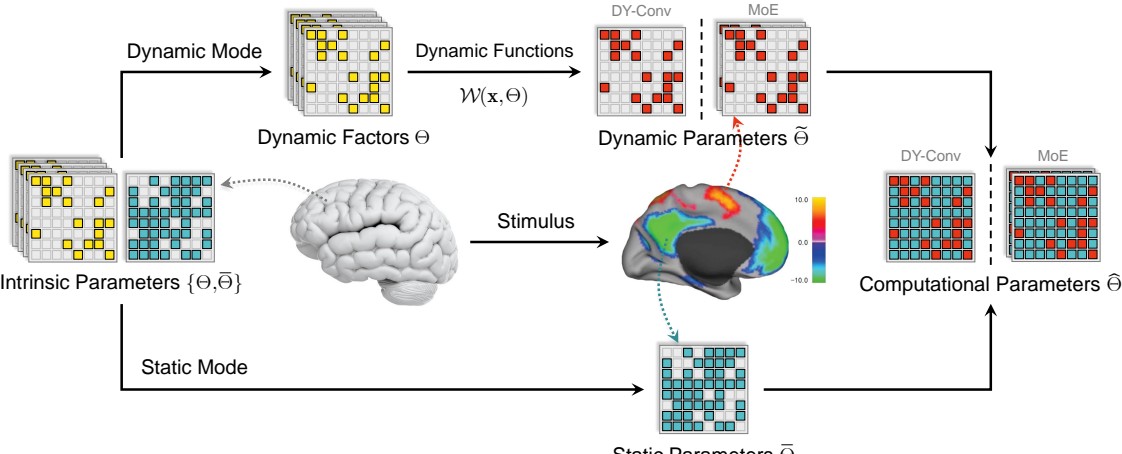

Figure 1: **The procedure of generating the computational parameters in PAD-Net, with DY-Conv and MoE as instantiations.** The intrinsic parameters include static parameters and dynamic factors. Given an input, dynamic factors activate and aggregate into dynamic parameters, which are then integrated with static parameters. As a result, computational parameters contain two modes of parameters, which is consistent with the map created through examination of spontaneous fluctuations in the functional MRI blood oxygen level-dependent signal in the human brain after a stimulus (Fox et al., 2005), where the task-positive network (warm color) activates while the task-negative network (cold color) remains the baseline state.

where $\hat{\Theta}_i(i = 1, 2, \ldots, m)$ represents the $i$-th element of $\hat{\Theta}$, and $\Theta$ denotes the dynamic factors. In our model, intrinsic parameters include dynamic factors $\Theta$ and static parameters $\bar{\Theta}$. Note that M partitions the computational parameters into two non-overlapping parts, forming a network with only a part of the parameters dynamic, i.e., Partially Dynamic Network (PAD-Net). Details of the procedure of generating the computational parameters from intrinsic are visualized in Figure 1.

According to Raichle et al. (2001), when doing cognitive tasks, the activity of the task-negative network and that of the task-positive network antagonizes each other, with one intense while the other weak (Cheng et al., 2020; Anticevic et al., 2012). Therefore, we set two scale factors to describe the intensity of these subnetworks separately, namely $\lambda_{\mathbf{s}}$ and $\lambda_{\mathbf{d}}$. With the above scale factors, our method can be factorized into a more general formulation:

$$\hat{\Theta}_i = \begin{cases} \lambda_{\mathbf{d}} \cdot \tilde{\Theta}_i & \text{if } M_i = 1 \\ \lambda_{\mathbf{s}} \cdot \bar{\Theta}_i & \text{otherwise} \end{cases}, \tag{4}$$

where we constrain $\lambda_{\mathbf{s}} + \lambda_{\mathbf{d}} = 2$ to simulate the antagonism effect, and Equation 3 is the special situation when both $\lambda_{\mathbf{s}}$ and $\lambda_{\mathbf{d}}$ are equal to 1. Similar to the constraint $\sum_{i=1}^{k} \pi_i$ in dynamic convolution, the constraint of summation compresses the parameters space and significantly simplifies the learning of $\lambda_{\mathbf{s}}$ and $\lambda_{\mathbf{d}}$ when joint optimizing scale factors and the counterpart parameters.

## 4.2 Identifying the Cherry on the Cake: Iterative Mode Partition

In the above section, we present the architecture of PAN-Net, which includes partly dynamic parameters and counterpart static parameters. In this part, we discuss how to generate the indicator mask that partition dynamic mode and static mode. Let us first formulate this partition as an optimization problem, where our goal is to find an appropriate indicator mask M to minimize loss $L$. Given a dataset $\mathcal{D} = \{(\mathbf{x}_i, \mathbf{y}_i)\}_{i=1}^{n}$ and a desired dynamic ratio $\kappa$ of M, we represent mode partition as the following constrained optimization

problem:

$$\min_{\mathrm{M}} L(\hat{\Theta}, \mathrm{M}; \mathcal{D}) = \min_{\mathrm{M}} \frac{1}{n} \sum_{i=1}^{n} \ell(\hat{\Theta}, \mathrm{M}; (\mathbf{x}_i, \mathbf{y}_i)),$$

$$\text{s.t.} \quad \mathrm{M} \in \{0,1\}^m, \quad \|\mathrm{M}\|_0 \leq \kappa \cdot m, \tag{5}$$

where $\ell(\cdot)$ denotes the standard loss function (e.g., cross-entropy loss), $\hat{\Theta}$ is the set of computational parameters of the neural network, $\|\cdot\|_0$ is the standard $L_0$ norm, $m$ is the total number of parameters. The conventional approach to optimize the above problem is adding sparsity enforcing penalty term to constrain the binary mask $\mathrm{M}$ (Carreira-Perpinán & Idelbayev, 2018), but it often requires heavily tuned hyperparameter settings and several trials. On the other hand, LTH-based (Chen et al., 2020a; Evci et al., 2020) methods can be borrowed to find the mask by several iterations, but it is prohibitively time-consuming. Also, considering the large model size of dynamic networks, the deployment of redundant parameters that will be pruned by $\hat{\Theta} \leftarrow \hat{\Theta} \odot \mathrm{M}$ and $\bar{\Theta} \leftarrow \bar{\Theta} \odot (1 - \mathrm{M})$.

We tend to partition the two modes before or early in training, abandoning the redundant parameters and avoiding time-consuming training iterations. Inspired by SNIP (Lee et al., 2019) that prunes the redundant parameters at initialization, we design an algorithm to tune excessive dynamic parameters into static mode before training. We resort to mini-batches of training data $\mathcal{D}_b = \{(\mathbf{x}_i, \mathbf{y}_i)\}_{i=1}^{b} \sim \mathcal{D}$ to detect redundant dynamic parameters. Specifically, if the $j$-th element in $\hat{\Theta}$ is dynamic, we can measure its importance of being dynamic by computing the loss difference $\Delta L_j$ caused by transforming $\hat{\Theta}_j$ into static one, which is represented by changing the value of $\mathrm{M}_j$ from 1 to 0:

$$\Delta L_j(\mathrm{M}, \hat{\Theta}; \mathcal{D}_b) = L(\mathrm{M}, \hat{\Theta}; \mathcal{D}_b) - L(\mathrm{M} - \mathbf{t}_j, \hat{\Theta}; \mathcal{D}_b), \tag{6}$$

where $\mathbf{t}_j$ is the indicator vector of $j$-th element of $\mathrm{M}$ (i.e., zeros everywhere except at the index $j$ where it is one). Note that we only consider transforming redundant dynamic parameters into static ones and the loss difference $\Delta L_j$ is zero if $\hat{\Theta}_j$ is already in static mode.

Note that computing $\Delta L_j$ for each dynamic parameter is prohibitively expensive, as it usually requires millions of forwarding pass over the dataset, so we resort to a simple and effective approximate alternative. Specifically, we release the binary constraints of $\mathrm{M}$ and make it differentiable and utilize the derivative of $L$ with respect to $\mathrm{M}_j$ to approximate $\Delta L_j$:

$$\Delta L_j(\mathrm{M}, \hat{\Theta}; \mathcal{D}_b) \approx g_j(\hat{\Theta}; \mathcal{D}_b) = \frac{\partial L_j(\mathrm{M}, \hat{\Theta}; \mathcal{D}_b)}{\partial \mathrm{M}} \bigg|_{\mathbf{t}=1} = \lim_{\delta \to 0} \frac{L_j(\mathrm{M}, \hat{\Theta}; \mathcal{D}_b) - L_j(\mathrm{M} - \delta \mathbf{t}_j, \hat{\Theta}; \mathcal{D}_b)}{\delta} \bigg|_{\mathbf{t}=1}, \tag{7}$$

where $g_j(\hat{\Theta}; \mathcal{D}_b)$ denotes the $j$-th element in derivative $g(\hat{\Theta}; \mathcal{D}_b)$. We accumulate the derivatives for all $j$ by one forward-backward pass using automatic differentiation. Note that if the magnitude of $g_j$ is high, it essentially means that transforming the parameter $\hat{\Theta}_j$ into static mode has a considerable effect on the loss, and it has to be preserved to maintain its dynamic mode. In contrast, the parameter should be transformed into static mode if the magnitude of $g_j$ is low. Therefore, We take the normalized magnitude of the derivatives of $g$ as the criteria:

$$s_j = \left| g_j(\hat{\Theta}; \mathcal{D}_b) \right| \Big/ \sum_{k=1}^{m} \left| g_k(\hat{\Theta}; \mathcal{D}_b) \right|. \tag{8}$$

Given the dynamic ratio $\kappa$, we take the $s_\kappa$ (the $\kappa$-th percentile of $s$) as the threshold and transform the mask elements whose scores are below into zero:

$$\mathrm{M} = \mathbb{1}\left[ s - s_\kappa \geq 0 \right], \tag{9}$$

where $\mathbb{1}[\cdot]$ is an element-wise indicator function where the output will be 1 if the condition $[\cdot]$ meets else it will be zero. Note that the indicator mask $\mathrm{M}$ prunes out redundant parameters in dynamic parameters $\tilde{\Theta}$ and static parameters $\bar{\Theta}$ respectively. Also, for fewer dynamic parameters to generate, we can also utilize the binary mask to prune redundant dynamic factors. Taking MoE as an example, $\mathrm{M}$ can be directly applied to parallel experts: $\Theta^{(i)} \leftarrow \mathrm{M} \odot \Theta^{(i)}, \forall i \in \{1, 2, \ldots, k\}$. In addition, we can decrease the computational cost of generating based on dynamic factors.

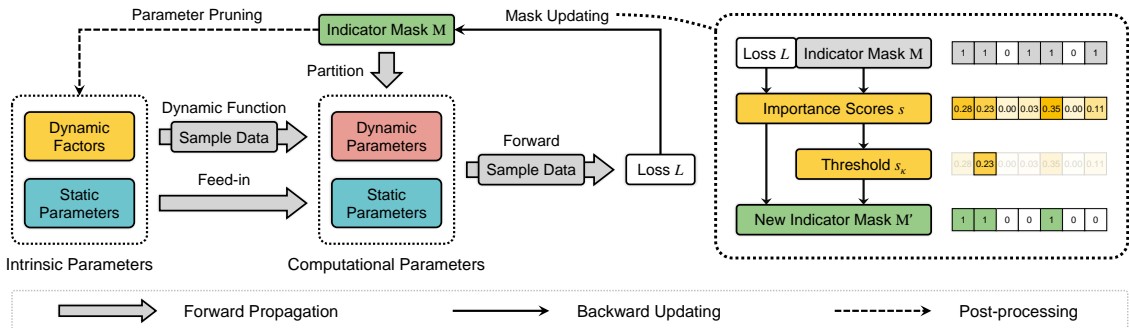

Figure 2: **Graphical illustration of Iterative Mode Partition (IMP).** *Left*: An overview of IMP, including forward propagation and backward updating. After IMP, the indicator mask prunes the redundant dynamic factors and static parameters (post-processing). *Right*: Details of mask updating.

Inspired by the success of iterative SNIP (Verdenius et al., 2020; de Jorge et al., 2021), we start from a fully dynamic network and adopt an iterative strategy shown in Figure 2 to transform dynamic parameters into static parameters step by step, where we increase the zero ratio of M exponentially. The effectiveness of the mode partition and the iterative mode partition are verified experimentally in Appendix D.

## 5 Empirical Evaluation

### 5.1 Main Results

**Visual Image Classification.** In Table 1, we compare our method to the static convolution (Krizhevsky et al., 2012), CondConv (Yang et al., 2019) and vanilla fully Dynamic Convolution (Chen et al., 2020b) on ImageNet (Deng et al., 2009) classification for ResNet (He et al., 2016) and MobileNetV2 (Sandler et al., 2018), by adjusting all convolution layers except the first layer. Before training, we first partition the two modes with a given dynamic ratio $\kappa$ using 10 batches of examples. Our method improves the classification performance with significantly lighter architecture and marginally fewer FLOPs (Floating Point Operations). For instance, ResNet-50 outperforms DY-Conv by 0.7% top-1 accuracy with 33.9% parameters and 0.1G fewer FLOPs. For more compact dynamic convolution variants, e.g., DCD (Li et al., 2021b) and ODConv Li et al. (2021a), we can see similar experimental results in Appendix C. The details of implementation are shown in Appendix A.

Table 1: **Comparison between PAD-Net and baselines for ResNet and MobileNetV2**, including CondConv and DY-Conv. The Top-1 accuracy is the averaged score for 5 runs, followed by the deviation. ✶ indicates the dynamic model with the fewest parameters or the fewer FLOPs (the static model is not included), and the best results in accuracy are **bold**. DY-Conv and PAD-Net contain $k = 4$ kernels, while CondConv contains $k = 8$ kernels.

| Depth | Model | Params | FLOPs | Top-1(w/dev) | Width | Model | Params | FLOPs | Top-1(w/dev) |
|---|---|---|---|---|---|---|---|---|---|
| ResNet-10 | Static | 5.2M | 0.89G | $63.1_{\pm0.4}$ | ×0.5 | Static | 2.0M | 97.0M | $65.7_{\pm0.3}$ |
| | CondConv | 36.7M | 0.92G | $66.9_{\pm0.2}$ | | CondConv | 15.5M | 113.0M | $68.8_{\pm0.2}$ |
| | DY-Conv | 18.6M | 0.91G | $67.4_{\pm0.3}$ | | DY-Conv | 4.0M | 101.4M | $69.6_{\pm0.1}$ |
| | PAD-Net | ✶**6.9M** | ✶**0.90G** | $\mathbf{68.1}_{\pm0.2}$ | | PAD-Net | ✶**2.7M** | ✶**98.3M** | $\mathbf{70.4}_{\pm0.2}$ |
| ResNet-18 | Static | 11.1M | 1.81G | $70.6_{\pm0.3}$ | ×0.75 | Static | 2.64M | 209.1M | $69.2_{\pm0.4}$ |
| | CondConv | 81.4M | 1.89G | $71.9_{\pm0.2}$ | | CondConv | 17.51M | 233.9M | $72.1_{\pm0.3}$ |
| | DY-Conv | 42.7M | 1.86G | $72.4_{\pm0.3}$ | | DY-Conv | 7.95M | 220.1M | $72.6_{\pm0.1}$ |
| | PAD-Net | ✶**15.1M** | ✶**1.83G** | $\mathbf{73.0}_{\pm0.3}$ | | PAD-Net | ✶**5.2M** | ✶**212.4M** | $\mathbf{73.5}_{\pm0.2}$ |
| ResNet-50 | Static | 23.5M | 3.86G | $76.2_{\pm0.2}$ | ×1.0 | Static | 3.5M | 300.8M | $72.1_{\pm0.3}$ |
| | CondConv | 129.9M | 3.98G | $76.9_{\pm0.3}$ | | CondConv | 27.5M | 329.0M | $74.4_{\pm0.2}$ |
| | DY-Conv | 100.9M | 3.97G | $77.2_{\pm0.2}$ | | DY-Conv | 11.1M | 312.9M | $74.8_{\pm0.2}$ |
| | PAD-Net | ✶**33.8M** | ✶**3.90G** | $\mathbf{77.9}_{\pm0.2}$ | | PAD-Net | ✶**6.1M** | ✶**304.4M** | $\mathbf{75.3}_{\pm0.1}$ |

**Natural Language Understanding.** We evaluate the performance of PAD-Net for MoE on various datasets from the General Language Understanding Evaluation (GLUE) benchmark (Wang et al., 2019), including linguistic acceptability (CoLA (Warstadt et al., 2019)), natural language inference (RTE (Bentivogli et al., 2009), QNLI (Rajpurkar et al., 2016), MNLI (Williams et al., 2018)), paraphrase and similarity (MRPC (Dolan & Brockett, 2005), STS-B (Cer et al., 2017), QQP (Iyer et al., 2017)), and sentiment classification (SST-2 (Socher et al., 2013)). Following the previous works (Lee et al., 2020; Dodge et al., 2020; Zhong et al., 2022b), we fine-tune the pretrained model (BERT (Devlin et al., 2019), ALBERT (Lan et al., 2020), RoBERTa (Liu et al., 2019), ELECTRA (Clark et al., 2020)) on the training set and directly report results on the validation set using the last checkpoint, since the test results are only accessible by the leaderboard with a limitation of the number of submissions.

Following the experimental setting in previous works (Shazeer et al., 2017; Gao et al., 2022), we replace the feed-forward layers with MoE layers where we prepare 8 experts and select the top-2 experts for each input. We set the dynamic ratio $\kappa = 50\%$ for it is close to the optimal ratio for performance. For more implemental details, please refer to Appendix A. Table 2 shows that PAD-Net outperforms MoE on the GLUE benchmark with a 0.95 average increase for four backbones. PAD-Net yields an average improvement of 1.9 percent for BERT and 0.7 percent for RoBERTa.

Table 2: **Comparison between PAD-Net and vanilla MoE** applied to four widely used large-scale Pretrained Language Models (PLMs). Averaged scores on all tasks are underlined. The shown results are the averaged score for 5 runs, followed by the deviation. The best results are **bold**. It shows that PAD-Net yields consistent improvements across all tasks among different MoE-equipped PLMs.

| Method | BERT | | | | | | ALBERT | | | | | |
|---|---|---|---|---|---|---|---|---|---|---|---|---|
| | #Param. | CoLA | RTE | MRPC | STS-B | Avg | #Param. | CoLA | RTE | MRPC | STS-B | Avg |
| Static | 103.3M | $54.6_{\pm0.4}$ | $66.4_{\pm0.7}$ | $84.6_{\pm0.3}$ | $85.8_{\pm0.3}$ | 72.9 | 11.1M | $54.2_{\pm0.7}$ | $76.6_{\pm0.7}$ | $87.2_{\pm0.4}$ | $90.6_{\pm0.3}$ | 77.2 |
| MoE | 346.9M | $58.0_{\pm0.9}$ | $69.3_{\pm1.2}$ | $85.0_{\pm0.4}$ | $87.1_{\pm0.2}$ | 74.9 | 29.2M | $56.8_{\pm1.2}$ | $77.2_{\pm0.8}$ | $87.4_{\pm0.4}$ | $90.7_{\pm0.3}$ | 78.0 |
| PAD-Net | **222.0M** | $\mathbf{59.7_{\pm0.8}}$ | $\mathbf{71.5_{\pm1.4}}$ | $\mathbf{85.5_{\pm0.4}}$ | $\mathbf{90.3_{\pm0.6}}$ | **76.8** | 21.3M | $\mathbf{57.4_{\pm1.4}}$ | $\mathbf{77.6_{\pm0.5}}$ | $\mathbf{88.4_{\pm0.3}}$ | $\mathbf{90.9_{\pm0.2}}$ | **78.6** |

| Method | RoBERTa | | | | | | ELECTRA | | | | | |
|---|---|---|---|---|---|---|---|---|---|---|---|---|
| | #Param. | CoLA | RTE | MRPC | STS-B | Avg | #Param. | CoLA | RTE | MRPC | STS-B | Avg |
| Static | 103.3M | $62.8_{\pm1.0}$ | $77.6_{\pm1.6}$ | $90.0_{\pm0.5}$ | $91.0_{\pm0.3}$ | 80.4 | 104.4M | $67.3_{\pm1.5}$ | $82.6_{\pm1.7}$ | $89.0_{\pm0.5}$ | $90.6_{\pm0.1}$ | 82.4 |
| MoE | 346.9M | $63.6_{\pm1.1}$ | $78.0_{\pm1.4}$ | $90.2_{\pm0.4}$ | $91.0_{\pm0.2}$ | 80.7 | 314.3M | $67.6_{\pm1.1}$ | $83.0_{\pm1.4}$ | $89.3_{\pm0.3}$ | $90.8_{\pm0.2}$ | 82.7 |
| PAD-Net | **222.0M** | $\mathbf{64.2_{\pm0.8}}$ | $\mathbf{79.4_{\pm1.2}}$ | $\mathbf{90.7_{\pm0.3}}$ | $\mathbf{91.4_{\pm0.3}}$ | **81.4** | 223.2M | $\mathbf{68.2_{\pm1.3}}$ | $\mathbf{84.1_{\pm1.5}}$ | $\mathbf{89.5_{\pm0.4}}$ | $\mathbf{91.2_{\pm0.2}}$ | **83.3** |

## 5.2 Ablation Study

**Dynamic Ratio.** Figure 3 shows the result of the ablation study on the ImageNet classification experiment for ResNet to investigate the impact of $\kappa$. Inspired by Wettig et al. (2022), we investigate the normalized performance[1] under a series of ratios (from 10% to 50%). Because PAD-Net with low dynamic ratios can outperform the fully dynamic networks, we thereby only consider ratios of less than 50%, which allows for more sparse and efficient models. As shown, we empirically find that $\kappa = 30\%$ is nearly the optimal ratio for vision dynamic models, e.g., ResNet, to achieve the highest performance, which is therefore left as the default setting. The abla-

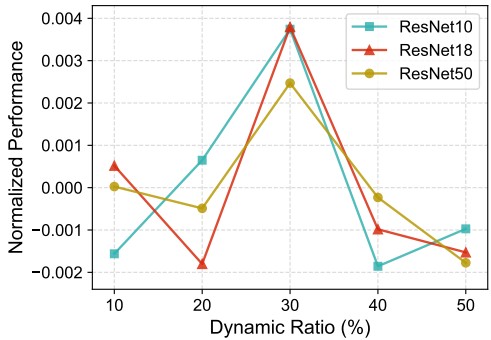

Figure 3: Impact of dynamic ratio on ResNet.

tion study for MoE can be seen in Appendix B, where the best performance is achieved when $\kappa = 50\%$. We believe that different dynamic functions have different optimal dynamic ratios, and an efficient way, e.g., hyper-parameter optimization or meta-learning, to search them will be necessarily explored in the future.

---

[1]Normalized performance is $\frac{x-\bar{x}}{\bar{x}}$ where $\bar{x}$ is the mean value of performance across all experiments.

**Scaled Factors.** Table 3 summarizes the influence of scale factors ($\lambda_s$ and $\lambda_d$) on different task performance. We initially tried to gain scale factors from a SENet structure (Hu et al., 2018), which did not contribute to the improvement of performance. Therefore we turn to set scale factors as trainable parameters to avoid redundant parameters and operations. In practice, besides the major setting "$\lambda_s + \lambda_d = 2$" to simulate the antagonism effect in Equation 4, we consider three other situations: only $\lambda_s$, only $\lambda_d$, and no scale factors. We conduct experiments on CIFAR-10 (Krizhevsky, 2009) and ImageNet for ResNet-50, RTE, and STS-B for BERT. As observed, $\lambda_s$ and $\lambda_d$ enhance performance substantially and their coexistence leads to the highest performance.

Table 3: The ablation study of scale factors.

| Model | Option | CIFAR-10 | ImageNet |
|---|---|---|---|
| | $-$ | 93.9 | 77.1 |
| | $\lambda_s$ | 94.3 | 77.2 |
| ResNet-50 | $\lambda_d$ | 94.5 | 77.4 |
| | $\lambda_s, \lambda_d$ | 96.0 | 77.6 |
| | $\lambda_s + \lambda_d = 2$ | **96.6** | **77.8** |

| Model | Option | RTE | STS-B |
|---|---|---|---|
| | $-$ | 69.6 | 87.4 |
| | $\lambda_s$ | 70.7 | 88.1 |
| BERT-base | $\lambda_d$ | 70.9 | 89.6 |
| | $\lambda_s, \lambda_d$ | 71.3 | 89.8 |
| | $\lambda_s + \lambda_d = 2$ | **71.5** | **90.3** |

To further validate the necessity of the summation constraint, i.e. 2, we release such constraint and test the setting "$\lambda_s, \lambda_d$". However, without the constraint of compressing the parameter space, our models encounter significant accuracy drops on ResNet-50 and BERT, i.e. -0.4 and -0.35, respectively.

### 5.3 Detailed Analysis

**The Difference with Model Pruning.** We compare mode partition with the model pruning method (Lee et al., 2019) on the GLUE benchmark for BERT-base and show the result in Table 4. Mode partition maintains important dynamic parameters while setting other redundant dynamic parameters in static mode, superior to pruning methods that directly zero out redundant parameters on the performance. Among all tasks of the GLUE benchmark, PAD-Net achieves the best performance, achieving a 1.1 higher average score than vanilla MoE. In contrast, we discover that the pruned MoE has decreased the performance significantly by 1.2 on the averaged score. Considering maintaining the performance improvement of a fully dynamic network, transforming redundant dynamic parameters into static ones is preferable.

Table 4: **Empirical comparison between our PAD-Net and model pruning on the GLEU benchmark**. The model pruning is set with $\kappa = 50\%$. We achieve the best performance, while the pruned model "MoE-P" suffers a performance drop compared to vanilla MoE.

| Models | #Param. | CoLA | SST-2 | MRPC | STS-B | QQP | MNLI | QNLI | RTE | Avg |
|---|---|---|---|---|---|---|---|---|---|---|
| Static | 103.3M | $54.6_{\pm0.4}$ | $91.4_{\pm0.3}$ | $84.6_{\pm0.3}$ | $85.8_{\pm0.3}$ | $90.6_{\pm0.2}$ | $83.7_{\pm0.4}$ | $90.4_{\pm0.2}$ | $66.4_{\pm0.7}$ | 81.2 |
| MoE | 346.9M | $58.0_{\pm0.9}$ | $91.7_{\pm0.2}$ | $85.0_{\pm0.4}$ | $87.1_{\pm0.2}$ | $90.8_{\pm0.1}$ | $83.8_{\pm0.2}$ | $90.4_{\pm0.1}$ | $69.3_{\pm1.2}$ | 82.0 |
| MoE-P | **222.0M** | $55.6_{\pm0.7}$ | $91.6_{\pm0.3}$ | $84.7_{\pm0.5}$ | $85.8_{\pm0.3}$ | $90.8_{\pm0.2}$ | $82.4_{\pm0.4}$ | $90.2_{\pm0.3}$ | $65.7_{\pm1.1}$ | 80.9 |
| PAD-Net | | $\mathbf{59.7}_{\pm0.8}$ | $\mathbf{92.2}_{\pm0.1}$ | $\mathbf{85.5}_{\pm0.4}$ | $\mathbf{90.3}_{\pm0.6}$ | $\mathbf{90.9}_{\pm0.1}$ | $\mathbf{84.2}_{\pm0.3}$ | $\mathbf{91.0}_{\pm0.1}$ | $\mathbf{71.5}_{\pm1.4}$ | **83.2** |

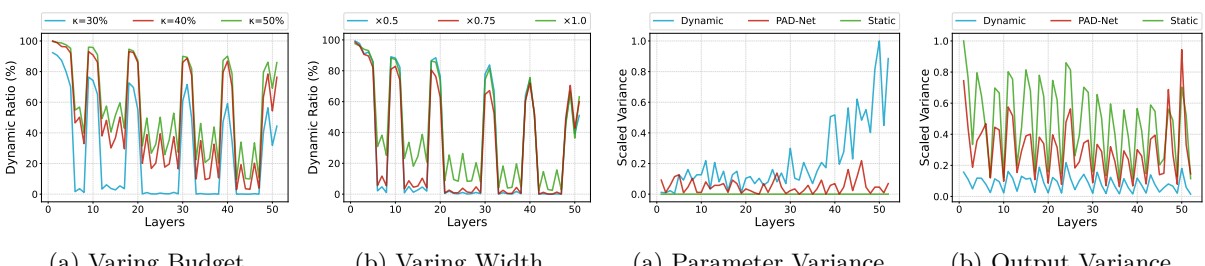

| (a) Varing Budget | (b) Varing Width | (a) Parameter Variance | (b) Output Variance |
|---|---|---|---|

| Figure 4: The layer-wise sparsity. | Figure 5: The dynamic property calculation. |
|---|---|

**Partially Dynamic Architecture.** We exert a flexible fine-grained strategy to partition two modes at the parameter level, leading to a layer-wise dynamic ratio distribution. We show the layer-wise dynamic ratio $\kappa$ distribution for Mobilenet-V2 in Figure 4a and 4b, where we consider two variables: the overall dynamic ratio and width multiplier. Figure 4 shows similar dynamic ratio distribution curves when changing the width

multiplier or overall dynamic ratio. We can notice that a large proportion of layers have a low dynamic ratio or even resemble static layers, which reveals the effectiveness of our proposed Cherry Hypothesis in dynamic convolutions. For MoE's dynamic ratio distribution and detailed architecture, we visualize the layer-wise indicator masks in Appendix E, which reflect the structured property of the distribution of parameters in different modes. Therefore, we believe IMP can be further applied to hardware-friendly structural methods.

**Dynamic Property.** Dynamic property refers to the variant numerical characteristics of a dynamic network fed by different inputs. The ideal dynamic network should require two capacities: assigning specific parameters for the input and making the output discriminating. Inspired by Li et al. (2021b), we take two levels of variance as metrics (parameter variance and output variance) to measure the dynamic property in Figure 5a and 5b. Static convolution, dynamic convolution, and PAD-Net ($\kappa = 30\%$) show different properties given the same test samples from ImageNet. We see that dynamic convolution retains a high degree of parameter flexibility, while its layer-wise outputs are less variable, while static convolution reveals the opposite phenomenon. The outputs of our method are more discriminating with fewer dynamic parameters, which may contribute to the superiority in performance and inform future work in dynamic network design.

## 6 Conclusion and Future Work

In this work, we first reveal the contradiction between the human brain and dynamic networks. To resolve this contradiction, we proposed the brain-inspired Cherry Hypothesis and assumed that a partially dynamic network could advance the performance and efficiency of fully dynamic networks. The proposed Partially Dynamic Network (PAD-Net) demonstrated the hypothesis in two frameworks - DY-Conv and MoE. Extensive experiments on both computer vision and natural language understanding tasks show the effectiveness and efficiency of PAD-Net against fully dynamic networks, which significantly improve the performance with much fewer dynamic parameters. Hopefully, our proposed method could be extensively integrated with other mainstream architectures and inspire future work in efficient neural network designation and other fields.

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

## A   Implementation Details

**Dynamic Convolution** We use an SGD optimizer (Ruder, 2016) with 0.9 momentum, following cosine learning rate scheduling and warmup strategy. The learning rate rises to the max learning rate linearly in the first ten epochs and schedules to arrive at zero within a single cosine cycle. We follow Chen et al. (2020b)'s temperature annealing strategy to avoid the unstable output values of the softmax function in the first epochs. We train the ResNet for 100 epochs with the max learning rate of 0.1. For the MobilenetV2, we train them for 300 epochs with the max learning rate of 0.05. The weight decay is 1e-4 for ResNet and 4e-5 for MobilenetV2. The training batch size is 256 for all models. To reduce variance, we shuffle the data with 5 random seeds and report both the average performance and deviation.

**Mixture of Experts** We use Adam (Kingma & Ba, 2015) as the optimizer with $\beta_1$, $\beta_2 = 0.9$, 0.98. For regularization, we set the weight decay as 0.1 and grid-search the learning rate from {1e-5, 2e-5, 5e-5, 1e-4, 2e-4}, where we warm up the learning rate in the first 10% steps (of the total training steps). For different

data scales, we grid-search the training epoch and batch size from {5, 10, 15, 20} and {8, 16, 32, 64}, respectively. The maximum length is 128 for all tasks. Following Shazeer et al. (2017), we first initialize the dynamic and static parameters with the pretrained parameters. The initial parameters are the same for all experts, we execute mode partition at the beginning of the second epoch when experts differentiate from each other.

## B Ablation Study on Dynamic Ratio for MoE

In Table 5, we investigate the effect of different dynamic ratios for MoE. We use BERT-base and RoBERTa-base as backbones. PAD-Net outperforms the fully dynamic MoE when $\kappa \geq 10\%$ and maintained stable performance when decreasing the dynamic ratio. The best performance is achieved when $\kappa = 50\%$. Considering the better performance, we set 50% as the default dynamic ratio for MoE in our work.

Table 5: **The ablation study for dynamic ratio on MoE integrated with PAD-Net.** Averaged scores on all tasks are underlined. The shown results are the averaged score for 5 runs. The best results are **bold**. Methods under the hdashline are our proposed PAD-Net, where $\kappa$ denotes the dynamic ratio.

| Method | #Param. | CoLA | SST-2 | MRPC | STS-B | QQP | MNLI | QNLI | RTE | Avg |
|---|---|---|---|---|---|---|---|---|---|---|
| BERT | 103.3M | 54.6 | 91.4 | 84.6 | 85.8 | 90.6 | 83.7 | 90.4 | 66.4 | 81.2 |
| w/ MoE | 346.9M | 58.0 | 91.7 | 85.0 | 87.1 | 90.8 | 83.8 | 90.8 | 69.3 | 82.1 |
| $\kappa = 70\%$ | 258.5M | 58.5 | **92.4** | **85.5** | 89.6 | 90.9 | 83.9 | 90.9 | 70.6 | 82.8 |
| $\kappa = 50\%$ | 222.0M | **59.7** | 92.2 | 85.4 | **90.3** | 90.9 | **84.2** | **91.0** | 71.5 | **83.2** |
| $\kappa = 30\%$ | 185.6M | 59.0 | 92.0 | 85.3 | 89.4 | **91.0** | 84.0 | 90.9 | 71.2 | 82.9 |
| $\kappa = 10\%$ | 149.1M | 57.5 | 92.1 | 85.4 | 88.3 | 90.7 | 84.1 | 90.6 | 70.2 | 82.4 |
| **Method** | **#Param.** | **CoLA** | **SST-2** | **MRPC** | **STS-B** | **QQP** | **MNLI** | **QNLI** | **RTE** | **Avg** |
| RoBERTa | 103.3M | 62.8 | 94.3 | 90.0 | 91.0 | 91.2 | 87.4 | 92.4 | 77.6 | 85.8 |
| w/ MoE | 346.9M | 63.6 | 94.8 | 90.2 | 91.0 | 91.7 | 87.7 | 92.9 | 78.0 | 86.2 |
| $\kappa = 70\%$ | 258.5M | 63.4 | 94.6 | 90.5 | 91.2 | 91.8 | 87.8 | **93.2** | 77.7 | 86.3 |
| $\kappa = 50\%$ | 222.0M | 64.2 | 94.4 | 90.7 | **91.4** | 91.8 | **87.9** | 93.0 | **79.4** | **86.6** |
| $\kappa = 30\%$ | 185.6M | **64.6** | 95.0 | **91.0** | 91.0 | **91.9** | 87.7 | 92.9 | 78.2 | 86.5 |
| $\kappa = 10\%$ | 149.1M | 63.9 | **95.2** | 90.4 | 90.9 | 90.9 | 87.6 | 92.7 | 78.8 | 86.3 |

## C Experimental Results of DCD and ODConv

We evaluate our method on DCD (Li et al., 2021b) and ODConv (Li et al., 2021a), more compact architectures for dynamic convolution. The results are shown in Table 6 and Table 7. We set the dynamic ratio 30% and use the same hyperparameter setting as proposed in the papers. The result shows that PAD-Net outperforms DCD and ODConv in both accuracy and efficiency, revealing that the combination of dynamic mode and static mode is better than the fully dynamic setting.

Table 6: Comparison between PAD-Net and DCD on ResNet and MobileNetV2.

| Depth | Model | Params | FLOPs | Top-1(w/dev) | Width | Model | Params | FLOPs | Top-1(w/dev) |
|---|---|---|---|---|---|---|---|---|---|
| ResNet-10 | DCD | 6.4M | 0.96G | $67.2_{\pm 0.4}$ | ×0.5 | DCD | 3.1M | 105.6M | $69.7_{\pm 0.2}$ |
| | PAD-Net | **⋆5.6M** | **⋆0.95G** | **$67.7_{\pm 0.2}$** | | PAD-Net | **⋆2.3M** | **⋆99.7M** | **$69.9_{\pm 0.3}$** |
| ResNet-18 | DCD | 14.7M | 1.84G | $72.6_{\pm 0.3}$ | ×0.75 | DCD | 4.1M | 222.9M | $72.4_{\pm 0.2}$ |
| | PAD-Net | **⋆12.2M** | **⋆1.82G** | **$73.0_{\pm 0.2}$** | | PAD-Net | **⋆3.1M** | **⋆213.2M** | **$72.8_{\pm 0.1}$** |
| ResNet-50 | DCD | 29.8M | 3.94G | $77.1_{\pm 0.2}$ | ×1.0 | DCD | 5.7M | 318.4M | $74.7_{\pm 0.4}$ |
| | PAD-Net | **⋆26.8M** | **⋆3.91G** | **$77.4_{\pm 0.1}$** | | PAD-Net | **⋆4.2M** | **⋆306.1M** | **$75.1_{\pm 0.3}$** |

Table 7: Comparison between PAD-Net and ODConv on ResNet and MobileNetV2.

| Depth | Model | Params | FLOPs | Top-1(w/dev) | Width | Model | Params | FLOPs | Top-1(w/dev) |
|-------|-------|--------|-------|--------------|-------|-------|--------|-------|--------------|
| ResNet-10 | ODConv(×1) | 5.5M | 0.90G | $67.6_{\pm0.2}$ | ×0.5 | ODConv(×1) | 2.4M | 101.8M | $69.1_{\pm0.4}$ |
| | PAD-Net(×1) | ✶5.4M | ✶0.89G | $\mathbf{67.9}_{\pm0.3}$ | | PAD-Net(×1) | ✶2.1M | ✶98.5M | $\mathbf{69.3}_{\pm0.3}$ |
| | ODConv(×4) | 19.7M | 0.93G | $68.0_{\pm0.3}$ | | ODConv(×4) | 4.4M | 106.4M | $70.3_{\pm0.2}$ |
| | PAD-Net(×4) | ✶ 9.7M | ✶0.92G | $\mathbf{68.2}_{\pm0.2}$ | | PAD-Net(×4) | ✶2.7M | ✶99.9M | $\mathbf{70.6}_{\pm0.2}$ |
| ResNet-18 | ODConv(×1) | 11.9M | 1.84G | $72.7_{\pm0.2}$ | ×0.75 | ODConv(×1) | 3.5M | 217.1M | $72.4_{\pm0.3}$ |
| | PAD-Net(×1) | ✶ 11.8M | ✶1.82G | $\mathbf{73.2}_{\pm0.3}$ | | PAD-Net(×1) | ✶2.9M | ✶211.5M | $\mathbf{72.8}_{\pm0.1}$ |
| | ODConv(×4) | 44.9M | 1.92G | $73.1_{\pm0.4}$ | | ODConv(×4) | 7.5M | 226.3M | $73.4_{\pm0.2}$ |
| | PAD-Net(×4) | ✶ 21.7M | ✶1.85G | $\mathbf{73.4}_{\pm0.2}$ | | PAD-Net(×4) | ✶4.1M | ✶214.3M | $\mathbf{73.6}_{\pm0.2}$ |
| ResNet-50 | ODConv(×1) | 28.6M | 3.92G | $77.6_{\pm0.2}$ | ×1.0 | ODConv(×1) | 4.9M | 311.8M | $74.7_{\pm0.2}$ |
| | PAD-Net(×1) | ✶26.5M | ✶3.89G | $\mathbf{77.9}_{\pm0.3}$ | | PAD-Net(×1) | ✶3.9M | ✶304.1M | $\mathbf{74.9}_{\pm0.1}$ |
| | ODConv(×4) | 90.7M | 4.08G | $78.0_{\pm0.2}$ | | ODConv(×4) | 11.5M | 327.1M | $75.4_{\pm0.3}$ |
| | PAD-Net(×4) | ✶44.2M | ✶3.93G | $\mathbf{78.2}_{\pm0.1}$ | | PAD-Net(×4) | ✶5.9M | ✶308.7M | $\mathbf{75.5}_{\pm0.2}$ |

## D The Effectiveness of Iterative Mode Partition

Figure 6 shows the effectiveness of mode partition and iterative mode partition, where we made a comparison among random partition "Random", mode partition "MP", dynamic partition "Dynamic", and iterative mode partition "IMP". Compared to fully dynamic networks, accuracy degrades when we partition the two modes randomly, which means this naive partition method mistakes some important dynamic parameters. In contrast, the mode partition that identifies the cherry on the cake contributes to a better architecture for higher performance. And the superiority of iterative mode partition demonstrates the effectiveness of the iterative strategy.

Figure 6: Comparison of different partition methods.

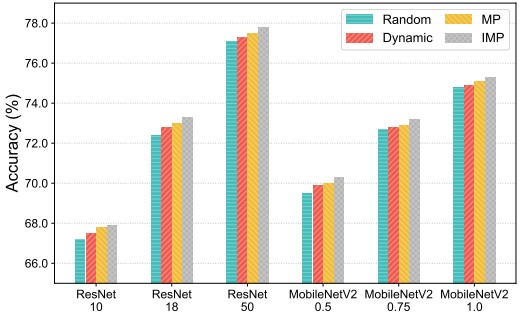

## E Mask Visualization of MoE

We visualize the 0/1 distribution of indicator masks and show the mode mapping for dynamic- and static-modes in Figure 7. Among layers, it reflects an incremental trend of dynamic ratios from bottom to top, which is consistent with the finding of Rücklé et al. (2021). Within a layer, we see a structured property, where the parameters of a column (or row) are mostly in the same mode. Given this, we guess partitions following a hardware-friendly structured manner may be close to the optimal, which informs future works.

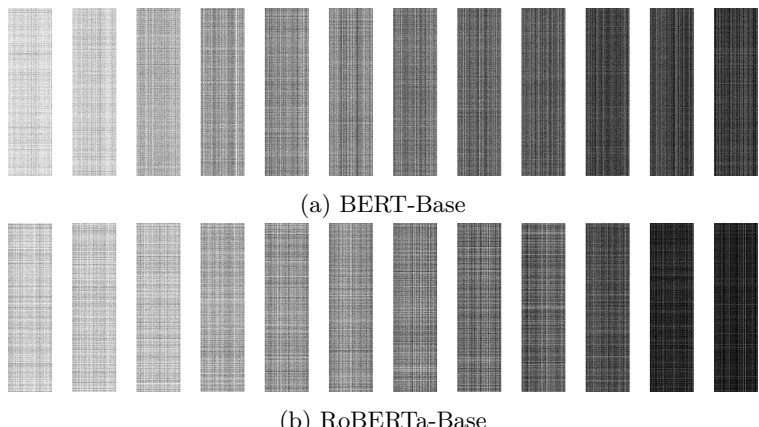

(a) BERT-Base

(b) RoBERTa-Base

Figure 7: Visualization of layer-wise indicator masks for MoE implemented in the feedforward layers of BERT-base and RoBERTa-base. We colored dynamic areas in black and static areas in white. From left to right, we display the masks from bottom layers to top layers and the dimension is $768 \times 3072$ for each mask.

