# OpenReview forum: "Cherry Hypothesis : Identifying the Cherry on the Cake for Dynamic Networks"
_TMLR — Withdrawn by Authors_

### Review · Reviewer_QjTX · 2022-11-28

**Summary Of Contributions:**

The paper advocates learning and deploying partially dynamic networks instead of fully dynamic ones to reduce unnecessary costs. It then develops a low-cost pruning method based on SNIP to achieve this. Experiments on both CV and NLP tasks provide validations for the proposal.

**Audience:**

Yes

**Claims And Evidence:**

No

**Requested Changes:**

Please address the raised issues in the Weaknesses part.

**Strengths And Weaknesses:**

# Strengths
- Reducing unnecessary dynamic parts of a dynamic model is well-motivated.
- The experiments are extensive.

# Weaknesses
- One major issue of this paper is that the "Cherry Hypothesis", which is the central concept in the abstract and introduction, has not been explicitly verified in the main text. I mean, although the results in Sec 5 can reflect this to some extent, it is better to deliver a systematic study on this hypothesis using various dynamic mechanisms, architectures, training recipes, etc., just like those in the "lottery ticket hypothesis" paper. In short, the current manuscript cannot convince me to accept the "Cherry Hypothesis" in general cases.

- The writing and presentation should be further improved. There are a few typos, even in some important expressions like Eq 7 (why is there a t=1 in the third term?) Sec 3 and Sec 4.1 should contain more technical details.

- The main technical contribution is the Iterative Mode Partition algorithm in Sec 4.2, and it is just a straightforward application of SNIP, which means the limited technical novelty of this paper. Although it has been argued repeatedly that novelty should not be a reason for rejecting a paper, I think the authors have not successfully revealed the unique challenges and technical innovations of applying SNIP here. Even if there are no challenges at all, the authors should clarify why they choose SNIP here among the pruning methods and provide more analyses and studies to convince the readers.

- Another major issue is that the authors perform unstructured pruning, and hence, though the FLOPS are reduced, the actual execution time is still high. Can you provide more (empirical) clarification on this?

- Regarding Table 1, I am just wondering, if the dynamic nets always rely on more parameters and more computations to achieve better performance than static nets, why can't we just deepen or enlarge those static nets to achieve similar performance? I am just curious about this as I am not familiar with this field.

---

### Review · Reviewer_5fQM · 2022-12-09

**Summary Of Contributions:**

The work considers dynamic neural networks, i.e., networks where the computational graph is conditional on the input itself. In particular, they focus on dynamic convolutions and mixture-of-experts (MoEs). They propose a "cherry hypothesis" (inspired by the lottery ticket hypothesis, LTH), stating that in a network with dynamic parameters we can identify a subset of parameters that can be turned to static values without hurting the performance. To identify this subnetwork, they extend SNIP by selecting parameters at initialization based on the gradient of the loss wrt a masking function. They evaluate the idea on two use cases (vision and NLP), showing small improvements in performance with a drastic reduction in memory for the parameters.

**Audience:**

Yes

**Broader Impact Concerns:**

I do not see a strong need for discussing broader impact here.

**Claims And Evidence:**

No

**Requested Changes:**

*) Memory cost: Do you also store the masks? Are you considering this cost in the experimental evaluation?

*) Clarify on what types of networks your setup can be applied (see point 3 above) and better clarify the motivation and the limitations (e.g., memory storage, computational cost).

*) Consider removing the hypothesis or renaming it based on my previous observations.

*) Eq. (4): you should add the constraint $\lambda_{\{\text{s},\text{d}\}} \ge 0$. From the text, I do not understand if the $\lambda$ coefficients are trained or not, and if they are trained, how are you handling the constraint? Finally, the ablation studying with no constraint is meaningless, because these are just coefficients multiplying the parameters (i.e., it's equivalent to having standard trainable parameters).

*) Consider reformulating all brain references (or possibly removing most of them). Below a brief list of sentences that in my opinion are misleading.
P1: "contradicts the previous discovery in the human brain [...]"
P1: "brain-inspired partially dynamic network"
P2: "these basic settings in dynamic networks contradict the phenomenon that neuroscience researchers observed in human brains"
P3: "Our motivation for adopting two subnets of parameters to fully dynamic networks is to borrow the functionality of TPN (while leaving the rest as TNN) in neuroscience."
P4: "Inspired by the co-working of task-negative networks (TNN) and task-positive networks (TPN)"
P5: "As a result, computational parameters contain two modes of parameters, which is consistent with the map created through examination of spontaneous fluctuations in the functional MRI blood oxygen level-dependent signal in the human brain after a stimulus"
P10: "In this work, we first reveal the contradiction between the human brain and dynamic networks."

*) Remove or rephrase in particular this sentence: "*Neural networks originate from the success of the human brain [?] in neuroscience. In this work, we first reveal the contradiction between the human brain and dynamic networks.*".

*) This sentence does not conclude (P6): "*Also, considering the large model size [...]*". In general there are some typos that should be fixed with some editing.

**Strengths And Weaknesses:**

Strengths: The topic of the paper (dynamic neural networks) is of increasing interest lately, which is a good point. The proposed Iterative Mode Partition (IMP) algorithm is a simple variation of SNIP but it is technically sound, and the experimental section supports (partially) the claims.

That said, I have several concerns on the paper which should be addressed or discussed.

1a) **RELATION TO NEUROSCIENCE**: first, while in the neural networks literature we are used to some vague claims on the neuroscientific basis of some algorithms (e.g., EWC in continual learning), it seems to me this is pushed to the extreme in this paper with inconsistent results. The paper claims inspiration from what are called task-negative network (TNN) and task-positive network (TPN), but this inspiration is dubious. A TNN is (supposed to be) a specific part of the brain that stays in a default mode when resting and deactivates when focusing on a task, contrary to the TPN. Here, the static parameters still contribute to the final prediction, and there is no "resting" state. Hence, I believe the analogy is flawed and I would suggest to strongly reduce these claims (see below for a full list).

1b) In addition, a brief glance at Wikipedia [1] reveals the TPN / TNN terminology is also out of date in neuroscience: "*This nomenclature is now widely considered misleading, because the network can be active in internal goal-oriented and conceptual cognitive tasks.*" Also [2]: "*These labels are more likely the byproduct of the desire for rigorously controlled experimental designs (i.e., externally directed stimuli) than meaningful descriptors of functional brain networks.*". This further validates my argument that neuroscientific claims should be kept to a minimum or possibly removed from the manuscript.

2) **NAMING AND MOTIVATION**: the LTH name is clear, since we can think of subnetworks at initialization as "winning tickets" due to their high performance. In addition, the LTH also addressed an open point in the literature at the time, namely, pruning was simple when training while challenging at the beginning. Finally, it was widely applicable to neural networks. In general, the name "*cherry*" does not have any clear relation to the problem under investigation, i.e., consider this sentence: "*Are all of them [parameters] cherries that lead to the promotion?*"

The framing of the hypothesis is also unclear: "*A fully dynamic network contains a subset of dynamic parameters that when transforming other dynamic parameters into static ones, can maintain or even exceed the performance of the original network.*" I think "*Other*" should be "*them [the subset]*" and the sentence should be reworded. In addition, the need for a named hypothesis is dubious, since a dynamic network in the sense here is obtained by multiplying the number of parameters of the original network, and it seems reasonable the performance should stay consistent when masking some of them. This is also applicable to a small set of techniques (see point 3 below). Overall, I believe authors should choose names that are more clear and possibly remove the emphasis on their "hypothesis".

3a) **DYNAMIC NETWORKS**: contrary to some claims in the paper, the method proposed here does not work on any dynamic NN, but only on networks where a parameter is selected among k options (either the filters of a convolution or the modules in a MoE). For example, it does not work for models with early exits (see Han et al., 2021). This should be clarified in the text. The authors also claim that the technique can reduce the computational cost (e.g., P6, "*In addition, we can decrease the computational cost of generating based on dynamic factors*"). Because the masking is sparse, it is unclear how much benefit we can gain from current hardware. For example, in a MoE, an expert should still be executed irrespective of whether 10%, 20%, or 50% of its parameters are "static" or "dynamic", and the routing is also agnostic to this. How can this model benefit from the techniques presented here? This is especially crucial since MoE are mostly studied for increasing the parameters for a fixed computational budget and/or to train in a distributed way efficiently, which is contrary to what is discussed here. This is reflected in the experiments, where the FLOPS differences are mostly small. This should be argumented better.

3b) I am also unsure about applying this method at initialization. While this makes sense for LTH (pruning), does it truly makes sense for the setup presented here? Routing, in particular, at the beginning is random, so it's unclear to me how much benefit we can get from the initial gradient. In my opinion an ablation study when considering the final trained network could be beneficial.

[1] https://en.wikipedia.org/wiki/Default_mode_network
[2] https://www.frontiersin.org/articles/10.3389/fpsyg.2012.00145/full

---

### Review · Reviewer_xRM1 · 2022-12-12

**Summary Of Contributions:**

The paper investigates whether a network with parameters that depend on the input (aka dynamic network) contains subsets of parameters that is best to keep static (ie. independent on the input). To decide which parameters to keep static, the paper introduces an approximate algorithm that during training searches which parameters is best to keep static. Results show a few percentage gain of accuracy by the proposed method in several benchmarks.

**Audience:**

Yes

**Claims And Evidence:**

No

**Requested Changes:**

- Connection with the brain: my suggestion is to remove this entirely as it is not needed and as it is now it is misleading.
- Hypothesis: it could also be entirely removed as it is not needed, the emphasis should be in the differences in the training algorithm.
- Training time: report it.
- Code: upload the code for the partially dynamic network and also add the readme that explain step by step how to run the code and reproduce the results.


Other questions:
- What does 'verify the cherry hypothesis' mean (page 2)? Please explain what is needed to 'verify'.
- The definition y = F(x, \theta | x) needs adjustment, as it is ambiguous where the term '| x' applies. It only applies to \theta, but as it is written it could apply to both x and \theta.
- The paper mentions several times that dynamic networks have 'redundant parameters and high deployment costs', but given the results it does not seem so, the differences are marginal.
- Why is the first layer excluded from the dynamic parameters (mentioned in page 7)?

**Strengths And Weaknesses:**

I think the paper is convincing in terms of improvement of accuracy in several benchmarks without scarifying much computational cost at test time. The paper is also clearly written and easy to follow.

The paper makes the following claims that are not accurate:
- The argument that the brain has static and dynamic parameters. While given an input and task some brain areas are active and others do not, this has nothing to do with the static and dynamic parameters in the paper. There is no evidence that the brain has parameters that are independent on the input and other parameters do not. Also, the term 'parameters' is misleading in this context, as the paper does not explain what parameters in the brain is referring to.
- The hypothesis that there is always a partially dynamic network better than a fully dynamic one is misleading. This is because a dynamic network can learn to leave some of the parameters static, ie. a dynamic network includes partially dynamic or even fully static. Thus, the main difference between the algorithm introduced in this paper and previous works is in the training algorithm of a dynamic network, in which some of the parameters are forced to be static, but note that the dynamic network can also lead to static parameters.
- The experimental results do not mention the overhead in training time to determine which parameters are forced to be static.

Finally, the code for the partially dynamic network is not included in the supplement. Please include it and also add a readme that explains how to reproduce the experiments.

---

### Note · Authors · 2022-12-23

I have read and agree with the venue's withdrawal policy on behalf of myself and my co-authors.